# Molecular characterization and antibiotic susceptibility of Shiga toxin- producing *Escherichia coli* (STEC) isolated from raw milk of dairy bovines in Khyber Pakhtunkhwa, Pakistan

Safir Ullah[1]*, Saeed Ul Hassan Khan[1]*, Tariq Ali[2], Muhammad Tariq Zeb[2], Muhammad Hasnain Riaz[2], Siraj Khan[3], Sagar M. Goyal[4]

1 Department of Zoology, Faculty of Biological Science, Quaid-i-Azam University, Islamabad, Pakistan, 2 Directorate of Livestock and Dairy Development Department Khyber Pakhtunkhwa, Veterinary Research Institute Peshawar, Peshawar, Pakistan, 3 Department of Pharmacy, Faculty of Biological Science, Quaid-i-Azam University Islamabad, Islamabad, Pakistan, 4 College of Veterinary Medicine, University of Minnesota, St. Paul, MN, United States of America

* drsafibm@yahoo.com (SU); saeedkhan@qau.edu.pk (SUHK)

**Data Availability Statement:** All relevant data are within the paper.

## Abstract

This study investigated the virulence potential and antibiotic susceptibility analysis of non-O157 Shiga toxin-producing *Escherichia coli* (STEC) serogroups, which are significant cause of food borne diseases. A study collected 800 samples of dairy bovine raw milk through various sources, 500 from milk shops, 200 from dairy farms, 26 from milk collection centers, and 74 from street vendors. Using a standard method, *E. coli* was detected in 321 out of the 800 samples collected. Out of the 321 *E. coli*-positive samples isolated, 148 were identified as STEC using selective media, specifically Cefixime Tellurite Sorbitol MacConkey's Agar (CT-SMA). Out of the 148 positive samples, 40 were confirmed as STEC non-O157 strains using multiplex PCR, indicating a prevalence of 5% (40 out of 800 samples). STEC isolates were subjected to antimicrobial susceptibility testing, and all isolates were resistant to at least one or more antimicrobials tested through the disk diffusion method, revealed high resistance to Amoxicillin 100%, Ceftriaxone 50%, and Penicillin 44.5%, and notably 44% of the strains exhibited Streptomycin resistance, while Enrofloxacin 55%, Florfenicol 50% and Norfloxacin 44%, demonstrated the highest susceptibility. Out of 40 STEC non-O157, twelve were subjected to Multi Locus Sequence Typing (MLST) sequencing through Illumina Inc. MiSeq platform's next-generation sequencing technology, United States. The genome investigation evidenced the persistence of twelve serotypes H4:O82, H30:O9a, H4:O82, H16:O187, H9:O9, H16:O113, H30:O9, H32:O, H32:O, H32, H32, and H38:O187, linked to the potential infections in humans. **Conclusion:** STEC isolates showed resistance to multiple antimicrobials, raising concerns for both animal and public health due to widespread use of these drugs in treatment and prevention. The study contributes new insights into monitoring STEC in raw milk, emphasizing the critical role of whole genome sequencing (WGS) for genotyping and sequencing diverse isolates. Still a deficiency in

**Funding:** The author(s) received no specific funding for this work.

**Competing interests:** The authors have declared that no competing interests exist

**Abbreviations:** STEC, Shiga toxin-producing *Escherichia coli*; mPCR, Multiplex polymerase chain reaction; ESBL, Extended-spectrum beta-lactamase; MLST, Multi Locus Sequence Typing; HUS, Hemolytic uremic syndrome; STEC/EHEC, Shiga toxin-producing, enterohemorrhagic; EPEC, Enteropathogenic; EAggEC, Enteroaggregative *E. coli*; DAEC, Diffusely adherent; HC, Hemorrhagic colitis; FAO, Food and agriculture organization; WGS, Whole genome sequencing; BPW, Buffered peptone water; EMB, Eosin Methylene Blue; VP, Vogues Proskuer; NSF, Non-sorbitol fermenting.

understanding STEC pathogenesis mechanisms, ongoing surveillance is crucial for safe-guarding human health and enhancing understanding of STEC genetic characteristics.

## 1. Introduction

Milk is an important source of nutrition for humans because it contains a wide range of essential nutrients. Its consumption is deeply rooted in tradition across both urban and rural regions of Pakistan [1]. Although raw milk is nutrient-rich, it is highly perishable and provides an optimal environment for the rapid growth of microorganisms. Consequently, it becomes a potential carrier for the transmission of foodborne illnesses [2]. An impactful zoonotic pathogen worldwide is Shiga toxin-producing *Escherichia coli* (STEC O157:H7). This Gram-negative bacterium is accountable for inducing severe health complications in those affected, leading to symptoms such as bloody diarrhea, hemorrhagic colitis, hemolytic uremic syndrome (HUS), and thrombotic thrombocytopenic purpura [3, 4]. *Escherichia coli* commonly inhabits the gastrointestinal tracts of both humans and animals, usually in a non-pathogenic form. It is crucial to recognize, that specific strains of *E. coli* can pose significant health risks to humans [5]. These pathogenic strains can be classified into various types, encompassing Shiga toxin-producing, enterohemorrhagic (STEC/EHEC), enteropathogenic (EPEC), enteroaggregative (EAEC), and diffusely adherent (DAEC) forms of *E. coli* [6]. Shiga toxin-producing *Escherichia coli* (STEC) is distinguished for generating multiple virulence factors, notably *stx1*, *stx2*, *eae*, and *ehxA*, which hold particular significance due to their cytotoxic effects. These factors contribute to conditions such as hemorrhagic colitis (HC) and hemolytic uremic syndrome (HUS). Ruminants, comprising cattle, buffalo, sheep, and goats are acknowledged as the primary reservoirs of STEC [7]. It is noteworthy that humans can act as reservoirs, contributing to person-to-person transmission, especially in familial environments, daycare centers, and institutional settings [8]. The primary source of STEC O157:H7 infection in humans is typically contaminated ground beef, and it's worth mentioning that contaminated food items such as unpasteurized milk and fruits or vegetables contaminated with feces can also be potential sources of infection [9]. Additionally, waterborne transmission of STEC has been documented, particularly after exposure to sewage-contaminated water, whether through swimming or drinking [10]. The occurrence of Shiga toxin-producing *E. coli* (STEC) in Pakistan has been documented in a variety of food products, with reported prevalence rates ranging from 1.94% to 56.4% [11]. Antibiotic drug resistance poses a significant challenge in global health [12]. The rising incidence of pathogenic multidrug-resistant (MDR) bacteria, notably extended-spectrum beta-lactamase (ESBL) and carbapenem-producing Enterobacteriaceae limits therapeutic options for clinicians and has prompted a renewed focus on older, potentially toxic drugs. [13]. Undoubtedly, the implementation of appropriate hygiene practices across the entire milk production chain, involving individuals such as milk producers, handlers, collectors, transporters, vendors, and consumers, is critical to safeguard the microbial safety of milk. Regrettably, in numerous developing countries, the adherence to hygienic practices is frequently deficient in the handling of milk and its derivatives, making unpasteurized or un-boiled milk potentially unsafe for consumption [14]. A study carried out by the Food and Agriculture Organization (FAO, 2018) disclosed significant annual economic losses across all stages of the dairy industry, spanning from production to consumption. In Pakistan, reports indicate the prevalence of Shiga toxin-producing *Escherichia coli* (STEC) in milk, ranging from 8% to 20%. [15]. The pathogenic nature of Shiga toxin-producing *Escherichia coli*

(STEC) is complex, necessitating an estimation of virulence beyond serotypes. Additional markers, such as virulence genes and phylogenetic indicators, should be considered. Whole genome sequencing (WGS) offers a comprehensive view of pathogen information, allowing for an assessment of the phylogenetic relationships between strains originating from diverse sources and geographic regions. The rising prevalence of antimicrobial-resistant *E. coli* poses a significant global public health concern [16]. Transfer of antimicrobial-resistant strains to humans can occur through the consumption of contaminated food. In Khyber Pakhtunkhwa, there is a lack of comprehensive reports on the characterization of Shiga toxin-producing *Escherichia coli* (STEC) in raw milk and dairy products. To address this gap, we conducted an analysis of isolated *E. coli* strains obtained from raw milk of dairy bovines from various sources. The study included antibiotic susceptibility testing, serotyping, and multilocus sequence typing (MLST) sequencing analysis. This research aims to provide technical support for investigations into the safety of dairy milk and to monitor the emergence of pathogenic STEC strains in the Khyber Pakhtunkhwa Province, Pakistan

## 2. Materials and methods

The research investigation took place across multiple institutions, including the Genomic Laboratory of Veterinary Research Institute (VRI), Peshawar (34.0170˚ N, 71.5699˚ E); the Laboratory of the Animal Science Institute at the National Agricultural Research Council in Islamabad, Pakistan 33.6995˚ N, 73.0363˚ E, and the College of Veterinary Medicine at the University of Minnesota in St. Paul, USA 46.7296˚ N, 94.6859˚ W.

### 2.1 Sample collection

In this study, a total of 800 samples were collected, including, 200 from dairy farms, 26 from milk collection points, 74 from street vendors, and 500 from milk shops across various regions of the Khyber Pakhtunkhwa Province during the period of 2020–2021, details mentioned in the (Fig 1). The analysis of *E. coli* was conducted using techniques that adhere to the standards recommended by the International Organization for Standardization (ISO) specifically designed for *E. coli* detection and characterization.

### 2.2 Pre-enrichment and bacterial isolation

In the analysis of raw milk samples, each 10 ml portion was individually mixed with 90 ml in a 100 ml sterile conical flask, and pre-enrichment medium (buffered peptone water—BPW) was added in 1:9 ratio, and processed under a laminar flow hood. The flask was sealed with cotton and wrapped in aluminum foil. Following this, samples were incubated overnight in a shaking incubator at 37˚C for duration of 24 h. A loopful of culture sample was streaked onto MacConkey agar plates and left to incubate at 37˚C for 24 h, resulting in the development of colonies with a pink color on the plates. Subsequently, the pink colonies from the MacConkey agar plates were sub-cultured on Eosin Methylene Blue (EMB) agar (Oxide, UK), where a characteristic metallic sheen was observed. The bacterial identity was confirmed through a series of tests, including Gram staining and biochemical tests such as Oxidase, Catalase, Indol, Methyl red, Vogues Proskauer (VP), and Citrate (IMVC). For the selective isolation of Shiga toxin-producing *Escherichia coli* (STEC), colonies were streaked on sorbitol MacConkey agar plates supplemented with 2.0μg/ml of cefotaxime (CT-SMAC). These plates were then incubated at 37˚C for 24 h, leading to the growth of white, colorless, and pink colonies mentioned in Table 1 and Fig 1.

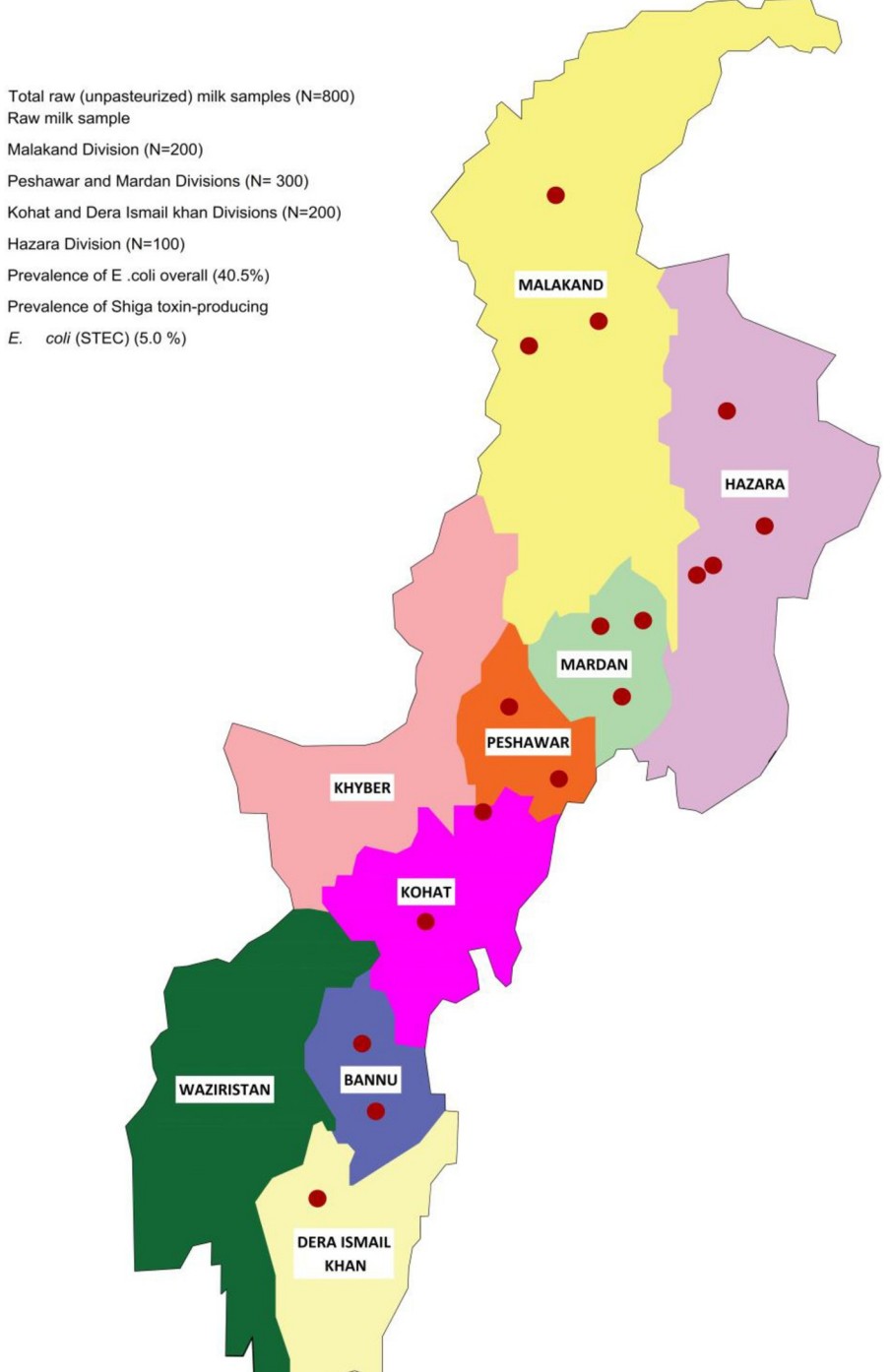

Total raw (unpasteurized) milk samples (N=800)
Raw milk sample

Malakand Division (N=200)

Peshawar and Mardan Divisions (N= 300)

Kohat and Dera Ismail khan Divisions (N=200)

Hazara Division (N=100)

Prevalence of E .coli overall (40.5%)

Prevalence of Shiga toxin-producing

*E.   coli* (STEC) (5.0 %)

**Fig 1. Geographical distribution of raw milk samples collected from dairy bovines in various regions of Khyber Pakhtunkhwa, Pakistan.**

## 2.3 DNA extraction

The 148 samples STEC-positive isolates from raw milk were selected for molecular characterization and confirmation of virulence genes through mPCR. Genomic DNA was extracted from *E. coli* (STEC) culture in brain heart infusion broth using Genomic DNA Extraction Kit

**Table 1. Cross tabulations of milk source with (*E.coli*, STEC non-O157:H7,) contaminations *n* = 800.**

| Source detail | *E.coli* | | STEC non-O157:H7 | | Total |
|---|---|---|---|---|---|
| | Positive | Not Detected | Positive | Not Detected | |
| Milk Shops | 207 | 293 | 30 | 470 | 500 |
| Dairy Farms and Individual Farmer | 073 | 127 | 08 | 192 | 200 |
| Milk Vendors | 028 | 046 | 01 | 073 | 074 |
| Milk Collection Centre | 013 | 013 | 01 | 025 | 026 |
| Total | 321 | 479 | 40 | 760 | 800 |
| Probability Statistics | p-Value = 0.449 | | p-Value = 0.305 | | |

(Invitrogen). Genomic DNA extraction was carried out from a 1ml sample of the overnight bacterial culture following the standard protocol provided by the kit. The specific measurement of DNA concentration in the extracted samples was determined using a Qubit 2.0 Fluorometer. The quality and reliability of the DNA extracted samples were loaded onto a 0.8% agarose gel, and electrophoresis was conducted. This gel electrophoresis process is a common method for assessing the reliability and purity of DNA samples.

## 2.4 Multiplex Polymerase Chain Reaction (mPCR)

The presence of encoding genes, namely *stx1*, *stx2*, *eae*, and *ehxA*, was determined through multiplex polymerase chain reaction (mPCR) assays. Genomic DNA, extracted from the samples in question, served as the template for these assays, utilizing primers specified in Table 2. The mPCR reaction mixture, with a total volume of 25μl, included 3μl of template DNA, 2.5μl of 10× PCR Buffer (Bio-Rad), 2μl of dNTP, 0.5μL of rTaq each for the forward and reverse primers, and 17.0 μl of nucleotide-free water [17]. The mPCR conditions involved an initial step at 95°C for 5 minutes, followed by 30 cycles of denaturation at 94°C for 30 seconds, annealing at various temperatures specified in Table 2 for 40 seconds, and extension at 72°C for 1 minute. The process concluded with a final extension at 72°C for 10 minutes before storage at 4°C. Positive and blank controls included DNA from the STEC O157:H7 strain and nucleotide-free water, respectively. Visualization of the mPCR product was achieved through 1% agarose gel electrophoresis under ultraviolet light.

**Table 2. Primers, concentration and annealing temperature used for molecular characterization of STEC O157: H7 (applied biosystems, Waltham, USA).**

| Genes | PCR conditions | PCR reaction volume |
|---|---|---|
| *Stx1, Stx2, eae, ehxA* | 1cycle | 2.5 μl of 10x PCR buffer |
| | 96°C-10 min | 0.15mM Mgcl$_2$ |
| | 35 cycles | 0.1mM of each dNTP |
| | 95°C-45 Sec | 0.5 μl of each primer |
| | 60°C-45 Sec | one unit of Taq DNA polymerase |
| | 72°C-45 Sec | 3 μl of DNA |
| | 1 Cycle final volume of 25 μl with sterile water 72°C -8 min | |
| 0157(*rfb* gene) | 1cycle 2.5 μl of | 2.5 μl of 10x PCR buffer 10x PCR |
| | 96°C-5 min | 0.15mM Mgcl$_2$ buffer |
| | 25 cycles | 0.1mMofeach dNTP (Thermoscientific,Waltham,USA |
| | 95°C-1min | 1.0 μl mM of 0157 *rfb* gene) primers |
| | 56°C, 58°C—1min | one unit of Taq DNA polymerase |
| | 72°C-1 min | 5 μl of DNA |
| | 1 Cycle,72°C -10 min held 4°C forever the final volume of 20 μl with sterile water | |

**Table 3. Antimicrobial susceptibility of shiga toxin-producing *E. coli* (STECnon-O157:H7) isolated from raw milk in Khyber Pakhtunkhwa.** (CLSI 2020).

| Antibiotics | | | | | | | |
|---|---|---|---|---|---|---|---|
| | Discs | Resistance | % | Intermediate | % | Susceptible | % |
| Penicillin | G p 10 iu | 8 | 44 | 6 | 33 | 4 | 22 |
| Amoxicillin | Aml 30 μg | 18 | 100 | 0 | 0 | 0 | 0 |
| Amoxicillin clavulanic acid | Aug 30 μg | 7 | 38 | 5 | 27 | 6 | 33 |
| Sulphathiazole | Smx 50 μg | 6 | 33 | 8 | 44 | 4 | 33 |
| Gentamicin | Cn 10 μg | 7 | 38 | 4 | 22 | 7 | 38 |
| Streptomycin | S 10 μg | 8 | 44 | 5 | 27 | 5 | 27 |
| Oxytetracycline | Ot 30 μg | 6 | 33 | 8 | 44 | 4 | 22 |
| Ceftriaxone | Cro 30 μg | 9 | 50 | 6 | 33 | 3 | 16 |
| Norfloxacin | Nor 10 μg | 6 | 33 | 4 | 22 | 8 | 44 |
| Enrofloxacin | Enr 5 μg | 5 | 27 | 3 | 16 | 10 | 55 |
| Florfenicol | Ffc 30 μg | 4 | 22 | 5 | 27 | 9 | 50 |
| Cefotaxime& clavulanic acid | Ctl 40 μg | 8 | 44 | 4 | 22 | 6 | 33 |
| Chi-Square = 37.01 | | | | P-Value = 0.023 | | | |

*RXC Method by using EPI-info software

## 2.5 Antibiotic susceptibility test

The antimicrobial susceptibility of the Shiga toxin-producing *Escherichia coli* (STEC) isolates was assessed using the disc diffusion method, following the protocols outlined by Kirby-Bauer and the Clinical and Laboratory Standards Institute. Each strain was plated on Muller Hinton agar seeded in three fields by cross streak and incubated at 37˚C for 24h. In order to verify the accuracy of this test 40 STEC strains were tested, adjusting to 0.5 of the McFarland [18]. It was then spread on Mueller-Hinton agar plates. After waiting for a few minutes, antibiotic discs were applied aseptically. A total of 12 types of antibiotics are available in the market. including Penicillin (p10 iu) Amoxicillin,(Aml30μg) Sulfamethoxazole (smx 50μg), Amoxicillin & Clavulanic acid,(aug30μg) Gentamicin,(cn10μg) Streptomycin,(s10μg) Oxytetracyclin,(ot30μg) Ceftriaxone (cro30μg), Norfloxacin, nor 10 μg Enrofloxacin, (enr 5μg) Florefenical, (ffc 30μg) and Cefotaxime & Clavulanic acid, (ctl 40μg) were employed in Table 3. The plates were then incubated for 24 h at 37˚C, and Measured the inhibition zone of diameter expressed the sensitivity of the bacteria, which defines the bacteria as resistant ($\leq$ 9 mm), moderately sensitive (10–11 mm), or sensitive ($\geq$ 12 mm) The results were interpreted by measuring the inhibition zone diameter and comparing those with the standard chart developed by Clinical Laboratory Standards Institute guidelines [19].

## 2.6 Whole genome sequencing assembly

The genomic DNA were extracted from 12 STEC strains and were processed for the sequencing using Illumina MiSeq platform's next-generation sequencing technology, following library preparation procedures provided by Illumina Inc., based at the University of Minnesota Genomic Center in the United States. Raw sequence data were analyzed using FASTAQ software, and subsequently, the reads underwent de novo assembly using the PATRIC (Pathosystems Resource Integration Center) web tool (https://www.patricbrc.org/). A filtration step was applied, excluding reads shorter than 150 base pairs. The resulting contigs were then annotated using Prokka, employing the default e value cutoff (version 1.13). It is important that for previously assembled genomes incorporated into the TORMS analysis, the same parameters were applied.

**Table 4. Serotyping status for STEC non-O157:H7 isolates from raw milk of dairy bovine In Khyber-Pakhtunkhwa.**

| Samples ID | H antigen | Fim H typing | O antigen |
|---|---|---|---|
| ST-202 | H4 | FimH type—*E.coli* | O82 |
| ST-203 | H30 | FimH type: fimH54 | O9 |
| ST-223-A | H4 | FimH type: fimH54 | O82 |
| ST-223-B | H4 | FimH type: fimH30 | O82 |
| ST-224 | H16 | FimH type: Unknown or presumptive new variant * | O187 |
| ST-228 | H9 | FimH type: fimH54 | O9a |
| ST 237 | H16 | FimH type: fimH53 | O113 |
| ST275 | H30 | FimH type: fimH54 | O9 |
| ST-289 | H9 | FimH type: fimH54 | O9a |
| ST-315 | H32 | FimH type—*E.coli* | - |
| ST-369 | H32 | FimH type—*E.coli* | - |
| ST-621 | H38 | FimH type: fimH54 | O187 |

## 2.7 *In silico* serotyping and Multi-locus Sequence Typing (MLST) analysis

The EC Typer tool for *Escherichia coli* Sero-typing, Version was used to confirm the serotype of isolates as O157 using default parameters O antigen minimum >90% identity coverage and H antigens minimum >90% identity and 60% coverage. Genetic relatedness of raw milk from dairy bovine strains were determined using an *in-silico E.coli* MLST Finder 2.0 scheme seven housekeeping genes loci adenylate kinase (*adk*), fumarate hydratase (*fumC*), DNA gyrase (*gyrB*), isocitrate/isopropylmalate dehydrogenase *(icd)*, malate dehydrogenase (*mdh*), *purA* (adenylosuccinate dehydrogenase), and *recA* (ATP/GTP binding motif) previously described for *E. coli* were used in MLST. The *E.coli* MLST database was used to assign a number to each locus and sequence type (ST) for each unique combination of loci [20]. As indicated in the Table 4.

## 3. Results

A total of (n = 800) raw milk samples were collected from dairy farms (200) milk collection, (74) street vendors (26) and milk shops (500) in different regions of Khyber Pakhtunkhwa, Province in 2020–21 (Fig 1) for PCR detection of Shiga toxin encoding genes (*stx*). Of the 800 samples (n = 800) tested, 5% (40 samples) were positive for *stx1*, *stx2*, *eae*, and *ehxA* virulence genes. Details were provided in Table 5 and Fig 2. The isolation rates of STEC strains were 4.0% (8/200) from dairy farms, 6% (30/500) from milk shops, 1.4% (1/74) from street vendors, and 3% (1/26) from milk collection centers. Detection of selected virulence genes (*stx1*, *stx2*,

**Table 5. Chi-square and fisher's exact test (statistics) of *E.coli* with (STEC non-O157:H7) *n* = 800.**

| *Escherichia coli* (*E.coli*) | Shiga Toxin producing *Escherichia coli* (STEC non-O157:H7) | | Total |
|---|---|---|---|
| | Positive | Not Detected | |
| Positive | 40 | 281 | 321 |
| Not Detected | 00 | 479 | 479 |
| Total | 40 | 760 | 800 |
| Chi-square = 62.830 | P-Value = 0.000 | Fisher's Exact Test = 0.000 | |

a. 0 cells (.0%) have expected count less than 5. The minimum expected count is 16.05.

b. Computed only for a 2x2 table

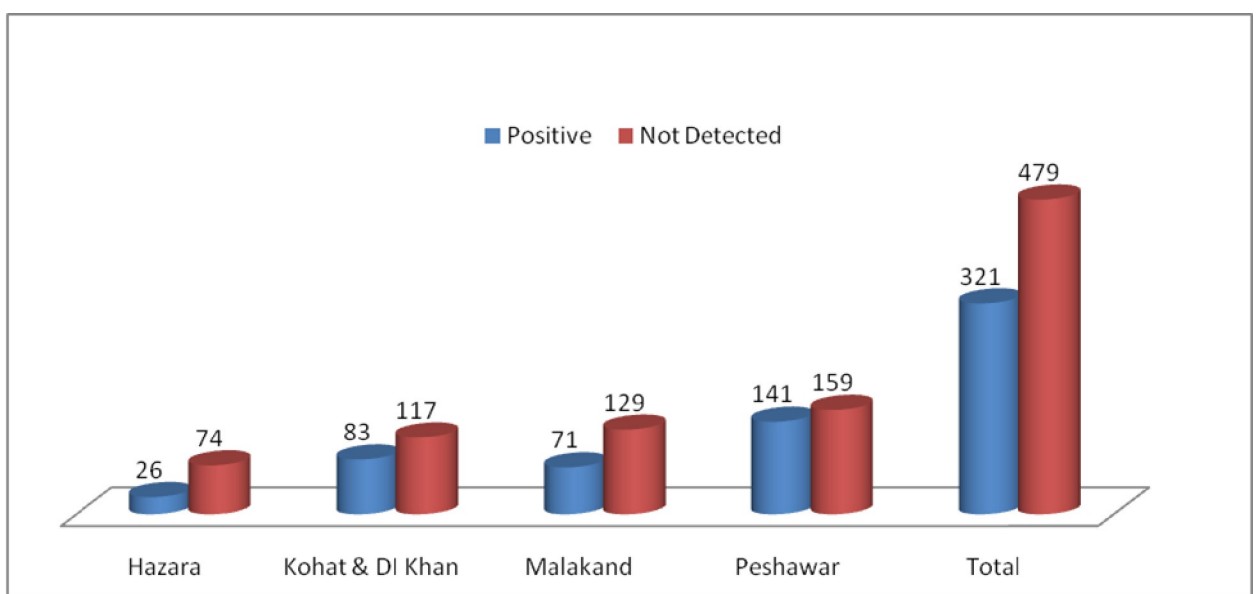

**Fig 2. Regional wise *E.coli* status *n* = 800.**

*eae*, *ehxA*) revealed that more than 5% (40/800) of the isolates carried more than two types of virulence-associated genes, specifically both *stx1* and *stx2* simultaneously (see Fig 3 and Table 6). In our study, analysis of raw milk isolates revealed that 38% of the isolates carried the *stx1* virulence gene, while 25% harbored the *stx2* gene (Fig 1). Additionally, the *hlyA* gene was detected in 19% of the raw milk isolates, Furthermore, all raw milk strains examined in this

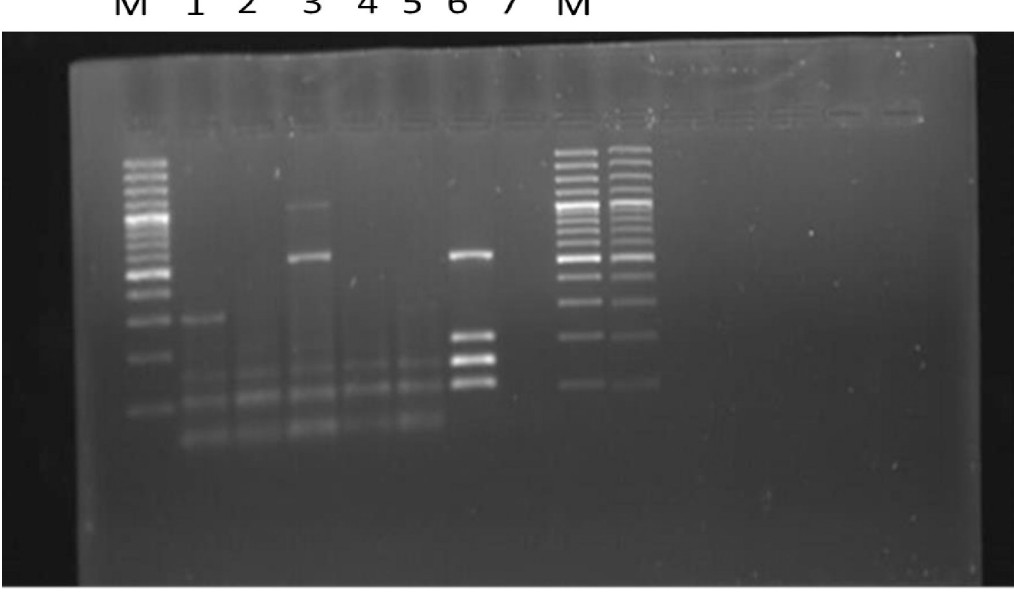

**Fig 3. Multiplex polymerase chain reaction (PCR) results for the presence of virulence genes (eae 100bp, *Stx*1 150bp, *Stx*2, *ehxA* 534bp) in Shiga toxin-producing *Escherichia coli* (STEC) strains isolated from raw bovine milk samples in Khyber Pakhtunkhwa, Pakistan.** The figure includes the following lanes: M Lane shows the 100bp ladder, Lane 1 is the positive control, Lane 2 is the negative control, and Lanes 3–6 represent the test samples.

Table 6. Prevalence of STEC O157:H7 virulence genes (*Stx1, Stx2, eae, ehxA*) from bovine raw milk.

| STEC Virulence genes | N (%) | Subtypes | Number of positive samples |
|---|---|---|---|
| *Stx1, eae* | 4 (10.0) | Not detected | 108(72.64%) |
| *Stx1, stx2, ehxA* | 5 12.5) | STEC non-O157 | 40 (5.0%) |
| *eae* | 5 (12.5) | | |
| *Stx1, ehxA* | 6 (15.0) | | |
| *ehxA* | 4 (10.0) | | |
| *Stx1, eae, ehxA* | 3 (7.5) | | |
| *Stx2, eae, ehxA* | 2 (5.0) | | |
| *Stx1, stx2* | 6 (15.0) | | |
| *Stx2* | 3(7.5) | | |
| *Stx1* | 2 (5.0) | | |
| Total | 40 (5.0%) | | 148 (18%) |

*Percentage W.R.T Total Number of Cases of 40 and Genes W.R.T STEC non-O157 Sub-Divisions.

study demonstrated the presence of the eae gene, accounting for 18% of the isolates (Fig 3). Additionally, all the STEC isolates were further subjected to serotyping via latex agglutination tests, using latex beads coated with specific antibodies STEC O157:H7 from Pro-Lax in the UK. Our data presented in the (Table 3) and result indicated that several antibiotics showed varying levels of effectiveness against the isolates. Enrofloxacin demonstrated the highest effectiveness, with 55% of isolates being susceptible. Similarly, Norfloxacin showed a high susceptibility rate of 50%, followed by Florfenicol at 44.5% and Gentamicin at 38%. In contrast, the isolates displayed low sensitivity to Ceftriaxone, with only 16% showing susceptibility. Amoxicillin on the other hand, exhibited complete resistance across all isolates, with 100% resistance observed. In our study, it was observed that all isolates exhibited resistance to gentamicin, Oxytetracyline, Amoxicillin and Clavulanic acid, and Cefotaxime. Antibiotic susceptibility profiles were comprehensively established for these isolates using twelve commonly used antibiotics from various drug classes (Table 3). Specifically, the isolates showed varying levels of resistance and susceptibility across the antibiotics tested. The results provide insight into the effectiveness of different antibiotic treatments against the studied isolates

Serotypes remain a crucial tool for epidemiological purposes as they allow for the characterization of lineages and the composition of populations. In this particular study, a total of 12 serogroups were identified among the STEC isolates, as shown in the Table 4.

### 3.1 *In Silico* sero-typing and multi-locus sequence typing

Forty samples bacterial isolates from raw milk of dairy bovine (n = 40) were checked and the quality of twelve samples DNA concentration found correct and were sequenced. In the *in-silico* serotyping revealed that all STEC non-O157 isolates possessed the H antigens determinant, and these serogroups, including H4:O82, H30:O9a, H4:O82, H16:O187, H9:O9, H16:O113, H30:O9, H32:O, H32-O, H32, H32 and H38:O187 were found respectively in Table 4, to be associated with the potential to cause infections in humans. MLST analysis revealed six different sequence types (ST). Most raw milk samples isolates belonged to a total of 12 categories of sequence types (ST) were identified within the 40 STEC isolates through the utilization of the MLST methodology (Fig 4). Phylogenetic analysis revealed that the MLST genotype ST 275, ST 224, and ST 223A, 203 exhibited a relationship and displayed a close affinity to epidemic MLST genotypes. Similarly, ST-223B, ST,-202, and ST-369, ST-315 showed a close relationship, among themselves. In contrast, ST-621 and ST237 exhibited distinct characteristics

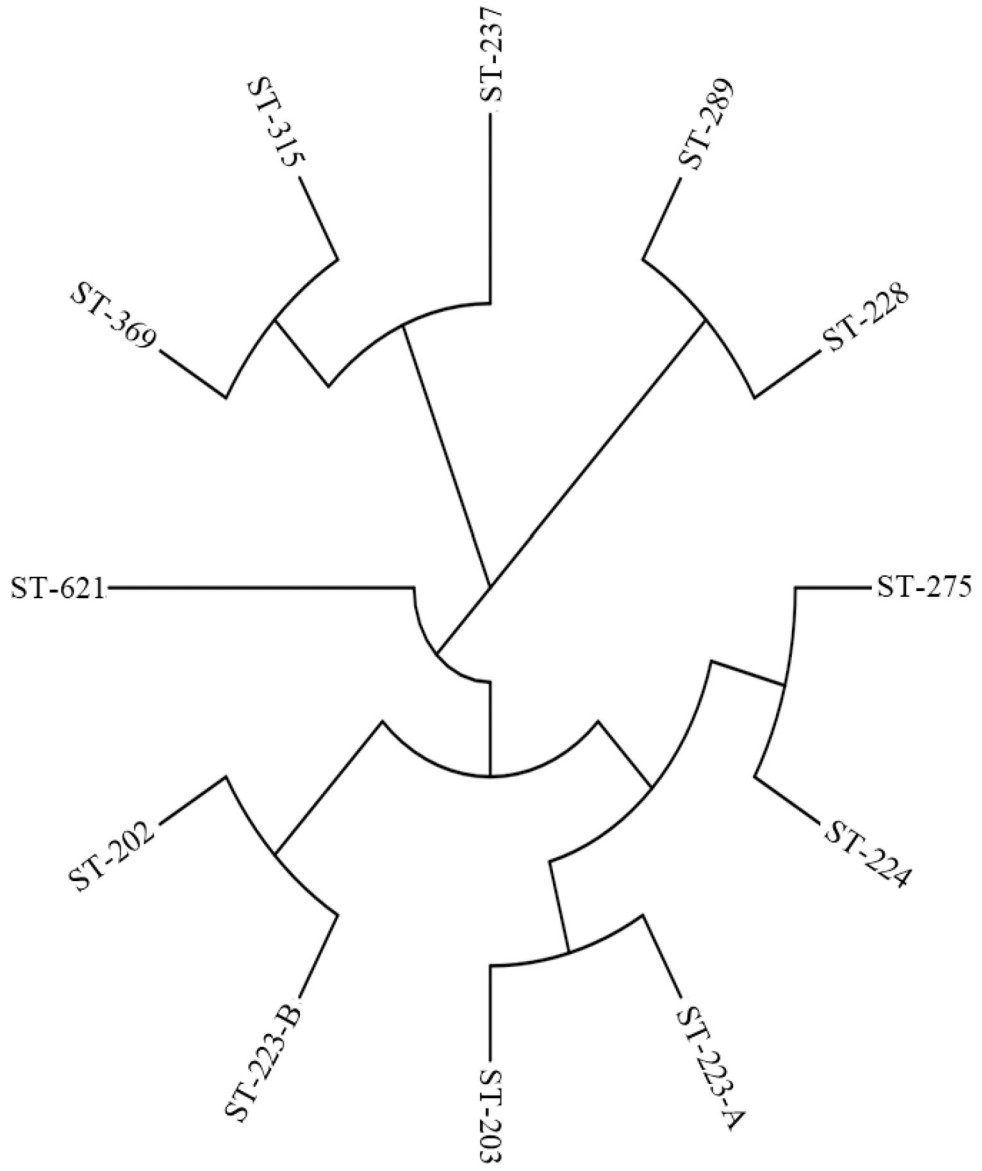

**Fig 4. MLST phylogenetic analysis.**

differing from the rest. In spirit, MLST has provided valuable insights into the genetic diversity and relationships among *E. coli* strains in a complex environment, shedding light on potential novel types and emphasizing the need for continued research in this field.

## 4. Discussion

Shiga Toxin-producing *E.coli* (STEC) are most infectious diarrheagenic *E coli* affecting the public health worldwide, STEC infections are also a leading cause of frequent foodborne illnesses [21]. Usually the most threatening STEC are those of O157:H7 serotype commonly result in hemolytic uremic syndrome (HUS), which is a life-threatening condition characterized by hemolytic anemia, and renal failure [22]. This study investigated the molecular characterization of STEC strains by determining the virulence genes and antibiotic resistance

profiles. These molecular factors play important roles in determining the pathogenicity of STEC, thus impacting public health. Out of the 800 raw milk samples, 321 (40.1%) were found to be positive for *E. coli.* following culturing on a specific medium, namely sorbitol MacCon-key agar, 148 (18.5%) of the positive samples were confirmed. Subsequently, when subjected to multiplex Polymerase Chain Reaction (mPCR) reaction protocol (Table 2). 40 (5.0%) of the isolates were identified as Shiga toxin-producing *E. coli*. These findings align with previous studies, such as the one [15], where 8.75% of milk samples from the city of Peshawar were positive for *E. coli* O157:H7, corresponding to our present results. Additionally, [23] reported an STEC prevalence of 5.7% in raw milk samples from the Apulia region. A study conducted [24]. found a prevalence of *E. coli* STEC O157:H7 at 10% in raw milk. Moreover, the serotype O157:H7 was identified in 6.21% of raw milk samples collected from cows, goats, and buffalos [25]. In the studied dairy bovine population, non-O157 STEC had a prevalence of 5.0% in raw milk. In contrast, the prevalence of this subgroup was 24.7% in Germany, 18.0% in Egypt, 58.1% in California, and 35.9% in Spain. In concordance to zero prevalence of serogroup O157 in this study, the low values were reported from other countries, zero % in Egypt and 3.8% in Spain. So, the dominance of non-O157 STEC subgroup over O157 is in agreement with [26], who reported the ranges of prevalence of nonO157 and O157 STEC from (0.42 to 74%) and (0.2 to 48.8%), respectively, around the globe. The variations in prevalence values might be due to the differences in epidemiological determinants like stocking density, age, season, spatial distribution, sampling time, strategy, handling and laboratory practices [27]. Another study conducted in Madurai, India, found *E. coli* in 65% of raw milk samples. The variation in the occurrence of *E. coli* in milk and milk products across different regions may be attributed to factors such as seasonal variations, farm size, feed type, hygiene practices, farm management, and differences in sampling procedures. The pathogenicity of STEC O157:H7 is linked to the production of Shiga toxin genes; including *stx*1, *stx*2, *eae*, and *ehxA*, or a combination of these toxins. To confirm the presence of Shiga toxin genes in the isolates, mPCR was carried out on all positive *E. coli* isolates as shown in the Fig 3. The results showed that 40 out of 148 (5%) isolates were positive for Shiga toxin genes (*stx*1, *stx*2, *eae*, and *ehxA*). These findings are consistent with [11], who reported 73% with s*tx*1 and 62% with *stx*2 in raw milk. In addition to stx1 and stx2, the prevalence of ehxA (19%) was also very high, showing a good agreement with previously studies. It is worthy of note that ehxA is generally used as a diagnostic indicator because the presence of ehxA is frequently correlated with the Shiga toxin [28]. In the present study, all eae-positive STEC strains isolated in Khyber Pakhtunkhwa, province was detected to be positive for stx2 in Table 6. The determination of such strains represents a high risk on public health in this region. However, it's important to note that results from various studies may not always align. For instance, some studies have reported a higher occurrence of *stx*2 compared to *stx*1 in bulk milk samples, and the presence of the *stx2* gene can vary in different regions, as indicated by a study on STEC O157 in beef cattle in Quetta, Pakistan [29]. The antimicrobial resistance (AMR) of STEC is also a serious problem that the world is now facing. It's worth noting that antibiotics such as Enrofloxacin (55%), Florfenicol (50%), Gentamicin (38%), and Norfloxacin (44%) demonstrated a notably high level of effectiveness in combating all the isolates. In contrast, the isolates exhibited a considerably low sensitivity to Ceftriaxone 16% and highly resistant Amoxicillin 100%. Additionally, the isolates displayed elevated levels of resistance against Penicillin, Amoxicillin, Clavulanic acid, Cefotaxime, Streptomycin, Oxytetracycline, and Sulfamethoxazole. In agreement with these studies, a low proportion of STEC isolates from the present study was susceptible to those types of antimicrobials mentioned in Table 3. These findings suggest a serious profile of AMR in STEC in food-producing animals. It is worthy of note that all STEC isolates were sensitive to Ciprofloxacin, Norfloxacin and Florfenicol in the present study. This raises concerns about their role as potential reservoirs for

multidrug-resistant STEC, which could potentially be transmitted to humans through the consumption of raw milk. This concern has also been reported in Lahore, Pakistan by [30]. The virulence potential of STEC, several factors should be considered, including serotype, *stx* subtype, virulence genes, phylogroups, and sequence type (ST). These strains belonged to different serotypes with various combinations of H and O antigens, including H4:O82, H9:O9a, H11:O160, H16:O113, H16:O156, H16:O187, H30:O9, H32, H32 and O187:H14 and H32:O mentioned in Table 4. They exhibited the ability to encourage infections in humans, substantiated by facts from the phylogenomic database of enteric bacteria. Certain subtypes of the *stx*2 gene (such as *stx2a*, *stx2c*, and *stx2d*) have been associated with severe human diseases. In this study, twelve isolates were found to possess these *stx2* subtypes, raising concerns about their potential to pose a health threat. *Stx*2 producing STEC strains, linked to patients with acute diarrhea and Hemolytic Uremic Syndrome (HUS), were also identified in this study, underscoring the clinical significance of the four isolates carrying *stx2*. In this study, 12 types of STs were determined for the 40 STEC isolates. In particularly, many isolates belonging to different STs possessed the same serogroups (Table 2). These findings are consistent with the findings. Suggesting that STEC isolates with the same serogroups might have genotypical diversity. Various STEC serotypes were identified in the raw milk samples, with O82:H4 and O9a:H9 being the most frequently detected serotypes. In another study conducted in American countries, STEC strains were found in cattle, beef products, and other food items [31]. Additionally, STEC O113:H21 strains were isolated from beef and cattle samples in Chile. Various sequence types (STs) of STEC strains have been isolated globally from both human and animal hosts, causing illnesses. For instance, MLST analysis of ST 203 and ST 223 revealed that H4-082 is associated with severe hemolytic uremic syndrome (HUS) infections. Similarly, non-O157 STEC lineages of H16: O187 have been linked to human infections in South American countries, with isolated cases reported in England. Different STEC strains have been reported in various countries, originating from environmental sources, food, clinical infections, and diverse animals [32]. Remarkably, several STs were identified, including, ST- 202, ST 203, ST 223, ST 224, ST 228, ST 237, ST 275, ST 369, ST 315, and ST 612 The ST prediction in this study identified the presence of the ST224 (O187:H14) clone in raw milk samples from Pakistan. Previous research has also reported the presence of STEC strains in cattle and among children. Additionally, STEC strains have been detected in food samples in South Korea, and they have been a predominant lineage in cattle samples in Sweden, as well as in pigs, cattle, milk, and water samples from dairy farms in China.

## 5. Conclusions

The non-O157 STEC serogroups with variable virulence characteristics were quite prevalent at the studied raw milk of dairy bovine in Khyber Pakhtunkhwa, Pakistan and because of its zoonotic potential may create public health hazards. It is crucial to monitor and regulate antibiotic usage in both animals and humans to curb the emergence of antibiotic resistance. Moreover, this data holds significance in gaining insights into the health implications associated with raw milk consumption and guiding future interventions to ensure the safety of dairy products in the region. Emphasizing milk pasteurization is recommended, along with the implementation of hygiene practices during milk production. Additionally, adopting appropriate herd management practices, eliminating high-shedders, and considering interventions such as vaccination, dietary adjustments, or the use of probiotics in feed are advisable measures. Finally, detailed studies are essential foe evaluating the Stx1 as a complete diagnostic tool for non-O157 serogroups across the globe.

## Acknowledgments

The authors thanks' the Veterinary Research Institute Peshawar, Animal Science Institute, National Agriculture Research Council Islamabad and University of Minnesota United States of America for providing research facilities and support for this study.

## Author Contributions

**Conceptualization:** Safir Ullah, Sagar M. Goyal.

**Data curation:** Siraj Khan.

**Investigation:** Safir Ullah, Sagar M. Goyal.

**Resources:** Tariq Ali, Muhammad Tariq Zeb, Muhammad Hasnain Riaz.

**Supervision:** Saeed Ul Hassan Khan.

**Validation:** Muhammad Hasnain Riaz.

**Writing – original draft:** Safir Ullah.

**Writing – review & editing:** Muhammad Hasnain Riaz, Siraj Khan.

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
