## [Decision Letter · Decision Letter 0]

1 Feb 2024

PONE-D-23-43641Molecular Characterization and Antibiotic Susceptibility of Shiga Toxin- Producing Escherichia Coli (STEC) Isolated from Raw Milk of Dairy Bovines in Khyber Pakhtunkhwa, PakistanPLOS ONE

Dear Dr. ullah,

Thank you for submitting your manuscript to PLOS ONE. After careful consideration, we feel that it has merit but does not fully meet PLOS ONE’s publication criteria as it currently stands. Therefore, we invite you to submit a revised version of the manuscript that addresses the points raised during the review process.

We look forward to receiving your revised manuscript.

Kind regards,

Sherin Reda Rouby, PhD

Academic Editor

PLOS ONE

Journal Requirements:

6. Please amend the manuscript submission data (via Edit Submission) to include author "Muhammad Hasnain Riaz". 

7. We note that Figure 4 in your submission contain map images which may be copyrighted. All PLOS content is published under the Creative Commons Attribution License (CC BY 4.0), which means that the manuscript, images, and Supporting Information files will be freely available online, and any third party is permitted to access, download, copy, distribute, and use these materials in any way, even commercially, with proper attribution. For these reasons, we cannot publish previously copyrighted maps or satellite images created using proprietary data, such as Google software (Google Maps, Street View, and Earth). For more information, see our copyright guidelines: http://journals.plos.org/plosone/s/licenses-and-copyright.

(1) You may seek permission from the original copyright holder of Figure 4 to publish the content specifically under the CC BY 4.0 license.  

Reviewers' comments:

Reviewer's Responses to Questions

**Comments to the Author**

1. Is the manuscript technically sound, and do the data support the conclusions?

Reviewer #1: Partly

Reviewer #2: No

2. Has the statistical analysis been performed appropriately and rigorously? 

Reviewer #1: N/A

Reviewer #2: N/A

3. Have the authors made all data underlying the findings in their manuscript fully available?

Reviewer #1: No

Reviewer #2: No

4. Is the manuscript presented in an intelligible fashion and written in standard English?

Reviewer #1: No

Reviewer #2: No

5. Review Comments to the Author

Reviewer #1: Line 33 Carried the virulence genes bla CTXM and bla TEM - These are antibiotic resistance genes

Line 36 - 37 Multi Locus Sequence Typing (MLST) was performed on twelve (12) - why only 12? why not all 40?

Line 55, 58, 61, 64, 70 - , as documented by (delete)

Line 63 - it's important to note

Line 78 - prevalence of STEC in Pakistan has been reported to vary significantly in various food products...

Line 79 - Antimicrobial resistance is seriously challenging the global health (rephrase this sentence)

Line 82 - Enterobacteriaceae (remove italics) rephrase sentence. It's not making sense.

Line 90 - as documented (FAO 2007) Use more recent refs.

Line 91 -Therefore forty isolated of these bacteria from raw milk from... (rephrase this sentence)

Line 109 - United States of America Materials and Methods

Line 119 - Indole

Line 120 - Voges Proskauer (VP)

Line 130 - Genomic DNA was extracted using the boiling method (17)

Line 146 - to perform an in silico molecular characterization of the sequenced strains. Please indicate which strains were sequenced and on which platform. Also, be clear on the bioinformatics tools that where used for analysis.

Line 163-164 - Unnecessary statement. Delete.

Line 167 - Kirby Bauer CLSI 2013. Why would you use guidelines from over 10 years ago? Please use more recent guidelines.

Line 172 - Antibiogram of the ESBL positive E. coli isolates was performed using antimicrobials (rephrase sentence)

Line 173, 178/179 - Put antibiotic disc concentrations

Line 184 - In Fig. 4

Line 184 - The study identified an E. coli prevalence rate of 40.5% among these samples are described in Table 1, and Fig:1 (rephrase this sentence)

Fig 2 - Very poor fig which is poorly annotated.

Fig 3 - poorly annotated figure. Say more about the presented isolates

Fig 4 - Poor figure. Put a key.

Table 2 - rfb gene mentioned for the first time in this table. The table needs fixing. PCR is a molecular technique, reagents are mixed at microlitre quantities not milliliters.

Table 3 - frequency of positive genes not adding up to 40.

Table 4 - Some antibiotic spellings are wrong e.g Sulphamathoxole, Oxytetracyclin, Ceftriexone. Put antibiotic concentrations. Why use Amoxy & AMC?

Line 257 - Define multi-drug resistance.

Line 267 - However, despite this constraint, the discovery of both bla TM and bla CTXM genes parallel is remarkable (rephrase the sentence). Why is the presence of these genes remarkable?

Reviewer #2: I regret to disappoint authors, but due to numerous methodological errors and ambiguities in the experiment described in the manuscript, I do not recommend this manuscript for publication.

- the authors' goal was to isolate and identify E. coli strains producing shiga toxin, i.e. STEC, from raw milk, because these are strains that may be responsible for various serious infections in humans.

I suggest changing the style in which the introduction was written a bit and removing sentence endings such as: as observed by, as highlighted, as discussed by, according to. They are unnecessary and make understanding difficult. Moreover, authors in such cases should refer to the names of the cited authors, not reference numbers

Lines: 99-102: goal description should be simplified; the sentence is too long and incomprehensible

Line 112: the authors performed selective pre-isolation in the presence of 0.1 mg/ml cefotaxime. According to the CLSI standard, the screening confirmatory test for ESBLs in E. coli uses a 100x lower concentration of cefotaxime. Therefore, with such selective multiplication, most E. coli strains will not grow - a serious methodological error that excludes further results obtained by the authors

Line 146: authors write about molecular characterization based on sequencing. No description of the sequencing procedure (methodology, which strains?)

Line 114: how to check turbidity in a cloudy, even diluted, milk sample?

Lines 157-163: no description of MLST procedure (only genes listed)

Antimicrobial susceptibility profiles:

The procedure for assessing drug susceptibility using the disc diffusion method is a standardized procedure, similar to the screening and confirmatory test for ESBL (the authors described a completely different methodology).

- antimicrobials used only in veterinary medicine were used, for which there are no criteria in the M100S standard. For some antibiotics used in human medicine there are no criteria in this standard either. So what criteria were used to assess drug susceptibility?

Lines 211-212: whether classical serotyping or molecular methods were used to determine serotypes and serogroups? The authors mix the two versions in “materials and methods” vs “results”

Lines217-221: please provide the specific STs obtained.

Due to the presented inconsistencies, the discussion and conclusion do not seem credible.

Minor comments:

Please use susceptible instead of sensitive and animicrobials instead of antibiotics.

Lines 93-95: I suggest deleting these sentences, the authors did not perform WGS

Lines 152-155: I suggest deleting it

6. PLOS authors have the option to publish the peer review history of their article (what does this mean?). If published, this will include your full peer review and any attached files.

Reviewer #1: **Yes: **Joshua Mbanga

Reviewer #2: **Yes: **Aneta Nowakiewicz

---

## [Author Response · Author response to Decision Letter 0]

2 May 2024

Thank you for sending reviewers’ comments on our manuscript on,

ONE-D-23-43641

Molecular Characterization and Antibiotic Susceptibility of Shiga Toxin- Producing Escherichia Coli (STEC) Isolated from Raw Milk of Dairy Bovines in Khyber Pakhtunkhwa, Pakistan

PLOS ONE “

We have revised our manuscript according to these comments. Answers to each of the comments are given below. I hope that this revised manuscript is now found suitable for publication, 

Reviewer 1 Comments highlighted in a (Red color)

 Answer. The manuscript has been completely revised in a proper PLOS one Style. The text presentation has been refined, inconsistencies have been eliminated, unity has been improved, and overall readability has been enhanced.

2. Please provide a complete Data Availability Statement in the submission form, 

Answer: Data availability statement submission form.

3. PLOS ONE now requires that authors provide the original uncropped and unadjusted images underlying all blot or gel results reported in a submission’s figures 

Answer: original uncroped gel results figures has been attached.

4. PLOS requires an ORCID ID for the corresponding 

Answer: ORCID iD is enlisted (0000-0002-96225217)

5. Please amend the manuscript submission data (via Edit Submission) to include author "Muhammad Hasnain Riaz". 

Answer: author name is included as per your suggestion.

6. We note that Figure 4 in your submission contain map images which may be copyrighted. All PLOS content is published under the Creative Commons Attribution License (CC BY 4.0), which means that the manuscript, images, and Supporting Information files will be freely available online

Answer: Figure 4 has been changed and new non map images attached..

5. Review Comments to the Author

Answer: needful done as per your suggestion.

Reviewer #1: Line 33 Carried the virulence genes bla CTXM and bla TEM - These are antibiotic resistance genes

Answer: needful done as per your suggestion.

Line 36 - 37 Multi Locus Sequence Typing (MLST) was performed on twelve (12) - why only 12? why not all 40?

Answer: MLST was performed on twelve isolates, for genome analysis. While the funding issue for genomic analysis.

Line 55, 58, 61, 64, 70 - , as documented by (delete)

Answer: as per suggestion needful done. 

Line 63 - it's important to note

Line 78 - prevalence of STEC in Pakistan has been reported to vary significantly in various food products..

Answer: needful done.

Line 79 - Antimicrobial resistance is seriously challenging the global health (rephrase this sentence)

Answer: rephrase has been done the sentences,

Line 82 - Enterobacteriaceae (remove italics) rephrase sentence. It's not making sense.

Line 90 - as documented (FAO 2007) Use more recent refs.

Line 91 -Therefore forty isolated of these bacteria from raw milk from... (Rephrase this sentence)

Line 109 - United States of America Materials and Methods

Line 119 - Indole

Line 120 - Voges Proskauer (VP)

Line 130 - Genomic DNA was extracted using the boiling method (17)

Line 146 - to perform an in silico molecular characterization of the sequenced strains. Please indicate which strains were sequenced and on which platform. Also, be clear on the bioinformatics tools that where used for analysis.

Line 163-164 - Unnecessary statement. Delete.

Line 167 - Kirby Bauer CLSI 2013. Why would you use guidelines from over 10 years ago? Please use more recent guidelines.

Line 172 - Antibiogram of the ESBL positive E. coli isolates was performed using antimicrobials (rephrase sentence)

Line 173, 178/179 - Put antibiotic disc concentrations

Line 184 - In Fig. 4

Line 184 - The study identified an E. coli prevalence rate of 40.5% among these samples are described in Table 1, and Fig:1 (rephrase this sentence)

Fig 2 - Very poor fig which is poorly annotated.

Fig 3 - poorly annotated figure. Say more about the presented isolates

Fig 4 - Poor figure. Put a key.

Table 2 - rfb gene mentioned for the first time in this table. The table needs fixing. PCR is a molecular technique, reagents are mixed at microlitre quantities not milliliters.

Table 3 - frequency of positive genes not adding up to 40.

Table 4 - Some antibiotic spellings are wrong e.g Sulphamathoxole, Oxytetracyclin, Ceftriexone. Put antibiotic concentrations. Why use Amoxy & AMC?

Line 257 - Define multi-drug resistance.

Line 267 - However, despite this constraint, the discovery of both bla TM and bla CTXM genes parallel is remarkable (rephrase the sentence). Why is the presence of these genes remarkable?

Answer: Your comments have been duly addressed and highlighted, and appropriately relocated to the material methods and results chapters after thorough revision.

Reviewer 2 Comments highlighted in a (purple color)

 I regret to disappoint authors, but due to numerous methodological errors and ambiguities in the experiment described in the manuscript, I do not recommend this manuscript for publication.

- the authors' goal was to isolate and identify E. coli strains producing shiga toxin, i.e. STEC, from raw milk, because these are strains that may be responsible for various serious infections in humans.

I suggest changing the style in which the introduction was written a bit and removing sentence endings such as: as observed by, as highlighted, as discussed by, according to. They are unnecessary and make understanding difficult. Moreover, authors in such cases should refer to the names of the cited authors, not reference numbers

Lines: 99-102: goal description should be simplified; the sentence is too long and incomprehensible

Answer: Both chapters have been revised and repetition has been removed, and strong links among sentences have been developed as per your comments. 

Line 112: the authors performed selective pre-isolation in the presence of 0.1 mg/ml cefotaxime. According to the CLSI standard, the screening confirmatory test for ESBLs in E. coli uses a 100x lower concentration of cefotaxime. Therefore, with such selective multiplication, most E. coli strains will not grow - a serious methodological error that excludes further results obtained by the authors.

Line 146: authors write about molecular characterization based on sequencing. No description of the sequencing procedure (methodology, which strains?)

Line 114: how to check turbidity in a cloudy, even diluted, milk sample?

Lines 157-163: no description of MLST procedure (only genes listed)

Antimicrobial susceptibility profiles:

The procedure for assessing drug susceptibility using the disc diffusion method is a standardized procedure, similar to the screening and confirmatory test for ESBL (the authors described a completely different methodology).

- antimicrobials used only in veterinary medicine were used, for which there are no criteria in the M100S standard. For some antibiotics used in human medicine there are no criteria in this standard either. So what criteria were used to assess drug susceptibility?

Lines 211-212: whether classical serotyping or molecular methods were used to determine serotypes and serogroups? The authors mix the two versions in “materials and methods” vs “results”

Lines217-221: please provide the specific STs obtained.

Due to the presented inconsistencies, the discussion and conclusion do not seem credible.

Answer. Changes have been made and followed the suggestion.

Minor comments:

Please use susceptible instead of sensitive and antimicrobials instead of antibiotics.

Lines 93-95: I suggest deleting these sentences, the authors did not perform WGS

Lines 152-155: I suggest deleting it

Answer. Changes has been made according to suggestion

---

## [Decision Letter · Decision Letter 1]

14 Jun 2024

PONE-D-23-43641R1Molecular Characterization and Antibiotic Susceptibility of Shiga Toxin- Producing Escherichia Coli (STEC) Isolated from Raw Milk of Dairy Bovines in Khyber Pakhtunkhwa, PakistanPLOS ONE

Dear Dr. ullah,

Thank you for submitting your manuscript to PLOS ONE. After careful consideration, we feel that it has merit but does not fully meet PLOS ONE’s publication criteria as it currently stands. Therefore, we invite you to submit a revised version of the manuscript that addresses the points raised during the review process.

ACADEMIC EDITOR:Please make an effort to address all comments by the reviewers as they are meant to improve the quality of your work. />==============================

We look forward to receiving your revised manuscript.

Kind regards,

Sherin Reda Rouby, PhD

Academic Editor

PLOS ONE

Additional Editor Comments :

Please make an effort to address all comments by the reviewer as they are meant to improve the quality of your work.

Reviewers' comments:

Reviewer's Responses to Questions

Comments to the Author

1. If the authors have adequately addressed your comments raised in a previous round of review and you feel that this manuscript is now acceptable for publication, you may indicate that here to bypass the “Comments to the Author” section, enter your conflict of interest statement in the “Confidential to Editor” section, and submit your "Accept" recommendation.

Reviewer #1: (No Response)

2. Is the manuscript technically sound, and do the data support the conclusions?

Reviewer #1: Partly

3. Has the statistical analysis been performed appropriately and rigorously? 

Reviewer #1: N/A

4. Have the authors made all data underlying the findings in their manuscript fully available?

Reviewer #1: No

5. Is the manuscript presented in an intelligible fashion and written in standard English?

Reviewer #1: No

6. Review Comments to the Author

Reviewer #1: Check paper for gramma especially abstract, but the whole paper needs to be checked and corrected.

In the initial manuscript there was mention of the ESBL genes bla CTX-M and bla-TEM but they’ve been removed in this revised version. Why? Was the detection of antibiotic resistance genes not done at all?

There was also mention of extracting DNA using the boiling method which has also been removed. Does this mean extraction was only done using the DNA kit mentioned in the revised version of the manuscript?

Always make an effort to address all comments by the reviewers as they are meant to improve the quality of your work. Wrong spellings e.g antibiotics, poorly annotated diagrams and weak grammar make the work less appealing. There’s need to improve on the technical aspects of the manuscript. The presentation of results needs improvement.

Line 28 Rephrase sentence. A total of 800 bovine raw milk samples from milk shops (500), ...

Line 30 write CT-SMA in full when writing for the first time.

Line 30 the positibve isolates were subjected to

Line 31 in silico

Line 32 rephrase sentence. 158 isolates were checked why are the percentages out of 800?

Line 34 say something about the listed serogroups

Line 37 ceftriaxone (correct spelling). Put percentages in brackets. Put a full stop after penicillin (44.5%).

Line 41 list the identified sequence types

Line 43 ...had the potential ability to transfer antibiotic resistance and virulence genes. This assertion is based on what exactly?

Line 132 the 0.5 MacFarland’s standard not 0.05

Line 138 indole, Voges Proskauer (VP)

Line 140 justify why you used the concentration of antibiotic you used. 0.1 gm of cefotaxime (CT-SMAC). Was it based on previous studies or was developed in this study.

Lines 144, 145 A subset of 40 samples matching to the same sampling years as the dairy bovine raw milk isolates was selected and subjected to sequencing. What do you mean by same sampling years?? Wasn’t selection based on the isolates that were positive in the STEC mPCR?

Lines 156 – 158 17µl Nuclease free water + 3µl DNA + 2.5µl PCR buffer + 1µl primers + 2µl dNTPs + 0.5µl Taq = 26µl in total. Please cross check and correct. Which enzyme (manufacturer, city) was used in the PCR?

Line 168 Put antibiotic disc concentrations

Line 171 and the zones... complete the sentence.

172 The CLSI document is an official document that can be cited independently. This must be done here.

Line 174 What was the selection criterion used to select the 12 isolates. This should be made clear.

Line 178 You need to put the web addresses of the tools you used for analysis e.g PATRIC (http://www.patricbrc.org). However PATRIC is now housed under BV-BRC (https://www.bv-brc.org/). Specify the specific tool/s you used for assembly, did you use SPAdes? Please provide all more detail on the bioinformatics analysis so that others may follow what you did.

Line 199 stx1, stx2, eae, ehxA. genes in table 2 and figure 2. Please correct this sentence

Line 200 and street vendors were and... Please correct this sentence

Line 202 100% (40/800) – this is not 100%

Line 204 All raw milk samples showed the presence of ... All means 100%. You can’t say all and then put a % which lower than 100%. Please correct this.

Lines 210 & 211 are the percentages susceptibility or resistance?

Line 212 ceftriaxone (16%) and were highly resistant to amoxycillin (100%)

Line 214 You’ve already mentioned the high resistance to amoxicillin or you meant to write amoxicillin and clavulanic acid (AMC)

Figure 3 – Poorly annotated. Lane 3 has pcr amplicons yet its said to be the negative control. Lanes 9 and 10 have the DNA ladder and can not be test samples. The figure casts doubts over the mPCR results presented in text.

Figure 4. Say something about the isolates in the legend. Multi locus sequence types typically are denoted by ST not SF.

Table 1 - rfb gene mentioned for the first time in this table.

Table 5 - Some antibiotic spellings are wrong e.g Sulphamathoxole, Oxytetracyclin, Ceftriexone.

7. PLOS authors have the option to publish the peer review history of their article (what does this mean?). If published, this will include your full peer review and any attached files.

Do you want your identity to be public for this peer review? For information about this choice, including consent withdrawal, please see our Privacy Policy.

Reviewer #1: Yes: Joshua Mbanga

---

## [Author Response · Author response to Decision Letter 1]

8 Jul 2024

Molecular Characterization and Antibiotic Susceptibility of Shiga Toxin- Producing Escherichia Coli (STEC) Isolated from Raw Milk of Dairy Bovines in Khyber Pakhtunkhwa, Pakistan PLOS ONE manuscript on, ONE-D-23-43641 

Thank you for sending reviewers’ comments on our manuscript.

We have revised our manuscript according to these comments. Answers to each of the comments are given below. I hope that this revised manuscript is now found suitable for publication.

S. # Comments Reply

 Comments to the author 

 1. If the authors have adequately addressed your comments raised in a previous round of review and you feel that this manuscript is now acceptable for publication, you may indicate that here to bypass the “Comments to the Author” section, enter your conflict of interest statement in the “Confidential to Editor” section, and submit your "Accept" recommendation. 

 Reviewer 1 (No response) Not Applicable (N/A)

 2. Is the manuscript technically sound, and do the data support the conclusions? The manuscript must describe a technically sound piece of scientific research with data that supports the conclusions. Experiments must have been conducted rigorously, with appropriate controls, replication, and sample sizes. The conclusions must be drawn appropriately based on the data presented. 

 Response to Reviewer: Thank you for the critical review and suggestions, which have been incorporated to enhance the quality of the manuscript to meet the esteemed journal's criteria. The experiments were conducted in various institutes as required, ensuring that all references are included in the manuscript. Adequate sample sizes and appropriate control measures were applied, and the results support the conclusions, making a significant contribution to the research database for further study. The revised version hopefully fulfills all criteria.

 3. Has the statistical analysis been performed appropriately and rigorously? 

 Reviewer #1: N/A N/A

 4. Have the authors made all data underlying the findings in their manuscript fully available?

 Response to Reviewer: Thank you for your review and suggestion. We have thoroughly revised the manuscript to adhere to the proper PLOS One style. The text has been refined for better presentation, inconsistencies have been addressed, unity has been improved, and overall readability has been enhanced. Additionally, the Data Availability Statement is now fully presented in the supplementary file, ensuring that all information is readily accessible without restrictions.

 5. Is the manuscript presented in an intelligible fashion and written in standard English?

 Response to Reviewer: Thank you for your critical review and comment. The revised manuscript has been carefully crafted to remain to the standard English format required by the journal. We have addressed language clarity and corrected grammatical errors during revision. We are confident that the manuscript now meets the publication criteria of the journal. 

 6. Review Comments to the Author

Response to Reviewer: Thank you for your critical review and comment. Regarding the research study, we assure you of its genuineness and the dual publication of its data. All ethical considerations have been meticulously addressed in accordance with the journal's criteria. Additionally, we have fulfilled all necessary requirements.

 Reviewer #1: Check paper for gramma especially abstract, but the whole paper needs to be checked and corrected.

In the initial manuscript there was mention of the ESBL genes bla CTX-M and bla-TEM but they’ve been removed in this revised version. Why? Was the detection of antibiotic resistance genes not done at all?

There was also mention of extracting DNA using the boiling method which has also been removed. Does this mean extraction was only done using the DNA kit mentioned in the revised version of the manuscript?

Always make an effort to address all comments by the reviewers as they are meant to improve the quality of your work. Wrong spellings e.g antibiotics, poorly annotated diagrams and weak grammar make the work less appealing. There’s need to improve on the technical aspects of the manuscript. The presentation of results needs improvement. 

Response to Reviewer: Thank you very much for the comments made in the manuscripts. I really appreciate your critical feedback. The ESBL genes blaCTX-M and blaTEM were initially mentioned, but due to lack of supportive data, we have removed them from the manuscript. Antibiotic susceptibility testing was performed instead of genotyping the resistant genes.

For isolation of DNA extraction for normal PCR is mostly done through the boiling process. However, for MLST, genomic DNA extraction was carried out using extraction kits. And DNA kit mentioned in the revised version of the manuscript.

We acknowledge the importance of addressing all reviewer comments to enhance the quality of our work. We addressed and corrected the spelling errors, tried to improve diagram annotations, and strengthen the grammar to make the manuscript more appealing. Additionally, we focused on enhancing the technical aspects and improving the presentation of our results.

Reviewer comments

Line 28 Rephrase sentence. A total of 800 bovine raw milk samples from milk shops (500), ...

Line 30 write CT-SMA in full when writing for the first time.

Line 30 the positive isolates were subjected to

Line 31 in silico

Line 32 rephrase sentence. 158 isolates were checked why are the percentages out of 800?

Line 34 say something about the listed serogroups

Line 37 ceftriaxone (correct spelling). Put percentages in brackets. Put a full stop after penicillin (44.5%).

Line 41 list the identified sequence types

Line 43 ...had the potential ability to transfer antibiotic resistance and virulence genes. This assertion is based on what exactly?

Line 132 the 0.5 MacFarland’s standard not 0.05

Line 138 indole, Voges Proskauer (VP)

Line 140 justify why you used the concentration of antibiotic you used. 0.1 gm of cefotaxime (CT-SMAC). Was it based on previous studies or was developed in this study.

Lines 144, 145 A subset of 40 samples matching to the same sampling years as the dairy bovine raw milk isolates was selected and subjected to sequencing. What do you mean by same sampling years?? Wasn’t selection based on the isolates that were positive in the STEC mPCR?

Lines 156 – 158 17µl Nuclease free water + 3µl DNA + 2.5µl PCR buffer + 1µl primers + 2µl dNTPs + 0.5µl Taq = 26µl in total. Please cross check and correct. Which enzyme (manufacturer, city) was used in the PCR?

Line 168 Put antibiotic disc concentrations

Line 171 and the zones... complete the sentence.

172 The CLSI document is an official document that can be cited independently. This must be done here.

Line 174 What was the selection criterion used to select the 12 isolates. This should be made clear.

Line 178 You need to put the web addresses of the tools you used for analysis e.g PATRIC (http://www.patricbrc.org). However PATRIC is now housed under BV-BRC (https://www.bv-brc.org/). Specify the specific tool/s you used for assembly, did you use SPAdes? Please provide all more detail on the bioinformatics analysis so that others may follow what you did.

Line 199 stx1, stx2, eae, ehxA. genes in table 2 and figure 2. Please correct this sentence

Line 200 and street vendors were and... Please correct this sentence

Line 202 100% (40/800) – this is not 100%

Line 204 All raw milk samples showed the presence of ... All means 100%. You can’t say all and then put a % which lower than 100%. Please correct this.

Lines 210 & 211 are the percentages susceptibility or resistance? Line 212 ceftriaxone (16%) and were highly resistant to amoxicillin (100%)

Line 214 You’ve already mentioned the high resistance to amoxicillin or you meant to write amoxicillin and clavulanic acid (AMC)

Figure 3 – Poorly annotated. Lane 3 has pcr amplicons yet its said to be the negative control. Lanes 9 and 10 have the DNA ladder and can not be test samples. The figure casts doubts over the mPCR results presented in text.

Figure 4. Say something about the isolates in the legend. Multi locus sequence types typically are denoted by ST not SF.

Table 1 - rfb gene mentioned for the first time in this table

Table 5 - Some antibiotic spellings are wrong e.g Sulphamathoxole

7. PLOS authors have the option to publish the peer review history of their article (what does this mean?). If published, this will include your full peer review and any attached files.

Do you want your identity to be public for this peer review? For information about this choice, including consent withdrawal, please see our Privacy Policy.

Response to reviewers:

The sentence has been rephrased. Please see line No. 28- 41 edited in red colored texts for ready reference please. 

Line 43 .The capacity to potentially transfer antibiotic resistance and virulence genes, examined through comprehensive genome sequencing and analysis. 

The sentence from line 132 has been corrected in lines 201-203 and the changes are marked in red for your quick reference.

The sentence from line 138 has been revised and corrected in lines 168-169. The changes are marked in red for your convenience."

The sentences from lines 144-145 have been changed, corrected, and highlighted in red. Please refer to lines 199-203 for the revised text.

The lines 156-158 have been revised. The concentration of PCR buffer and primers used has been corrected and highlighted in red text on lines 210-213 for easy reference.

The antibiotic disc concentrations have been added to line 168 of the manuscript. Please refer to lines 230-232 for the revised text.

Line 171 has been revised with corrections, and the description of the zones of inhibition is now included in the sentence. Please refer to lines 204-214 for details. The CLSI document references for line 172 have been documented. Please see line 234 for details.

 Line 174 has been changed, and the criteria for antibiotics include a total of 12 types available in the market. Please refer to lines 231-232 for the text highlighted in red.

 Line 178 has been updated with corrections and includes detailed information on next-generation sequencing (NGS) technology using the Illumina MiSeq platform by Illumina Inc., conducted at the University of Minnesota Genomic Center in the United States. Raw sequence data was processed using FASTAQ software, followed by de novo assembly using the PATRIC web tool and annotation with Prokka (version 1.13). Please refer to lines 220-221 for more information.

Line 199 has been revised with corrections. Please refer to lines 240-241 for the text highlighted in red for your reference.

The sentence at line 200 concerning street vendors has been corrected. Please refer to lines 242-244 for highlighted text for your convenience.

The lines 202 to 204 have been changed, sentences have been revised and corrected, and please refer to lines 246-252 for highlighted text for ready reference.

Lines 210 & 211 have been revised and edited. Please refer to lines 259-270 for ready reference

Line 214 have been revised and changed amoxicillin and Clavulanic acid (AMC) see line 266 for ready reference.

, Oxytetracyclines, Ceftriexone. Thank you very much for the comment. 

Figure 3 has been updated with properly marked of PCR amplicons. Please see figure no 3.

Figure 4: MLST designated the sample sequence type as ST, but the samples were marked as SF during Samples library preparation and laboratory identification for whole genome sequencing.

Table 1 includes the rfb gene, which was part of our research for identifying STEC O157 strains, but no findings were made. Therefore, its inclusion was necessary.

Table 5 The spelling of antibiotics has been corrected. Please refer to it for references,

Thank you for your option to publish the peer review history of this article. I opt to publish the peer review history of my article. Please proceed accordingly.

---

## [Editor Report · Decision Letter 2]

12 Jul 2024

Molecular Characterization and Antibiotic Susceptibility of Shiga Toxin- Producing Escherichia Coli (STEC) Isolated from Raw Milk of Dairy Bovines in Khyber Pakhtunkhwa, Pakistan

PONE-D-23-43641R2

Dear Dr. ullah,

We’re pleased to inform you that your manuscript has been judged scientifically suitable for publication and will be formally accepted for publication once it meets all outstanding technical requirements.

Kind regards,

Sherin Reda Rouby, PhD

Academic Editor

PLOS ONE
---

## [Editor Report · Acceptance letter]

22 Jul 2024

PONE-D-23-43641R2 

PLOS ONE

Dear Dr. ullah, 

I'm pleased to inform you that your manuscript has been deemed suitable for publication in PLOS ONE. Congratulations! Your manuscript is now being handed over to our production team.

Kind regards, 

on behalf of

Professor Sherin Reda Rouby 

Academic Editor

PLOS ONE